# Dynamic Multi-Team Racing: Competitive Driving on 1/10-th Scale Vehicles via Learning in Simulation

**Peter Werner**[*,1], **Tim Seyde**[*,1], **Paul Drews**[2], **Thomas Balch**[2], **Wilko Schwarting**,
**Igor Gilitschenski**[3], **Guy Rossman**[2], **Sertac Karaman**[4], **Daniela Rus**[1]

[1] MIT CSAIL, [2] Toyota Research Institute, [3] UofT RI, [4] MIT LIDS, [*] equal contribution
{wernerpe,tseyde}@mit.edu

**Abstract:** Autonomous racing is a challenging task that requires vehicle handling at the dynamic limits of friction. While single-agent scenarios like Time Trials are solved competitively with classical model-based or model-free feedback control, multi-agent wheel-to-wheel racing poses several challenges including planning over unknown opponent intentions as well as negotiating interactions under dynamic constraints. We propose to address these challenges via a learning-based approach that effectively combines model-based techniques, massively parallel simulation, and self-play reinforcement learning to enable zero-shot sim-to-real transfer of highly dynamic policies. We deploy our algorithm in wheel-to-wheel multi-agent races on scale hardware to demonstrate the efficacy of our approach. Further details and videos can be found on the project website: https://sites.google.com/view/dynmutr/home.

**Keywords:** Multi-Agent RL, Autonomous Racing, Sim-to-Real Transfer

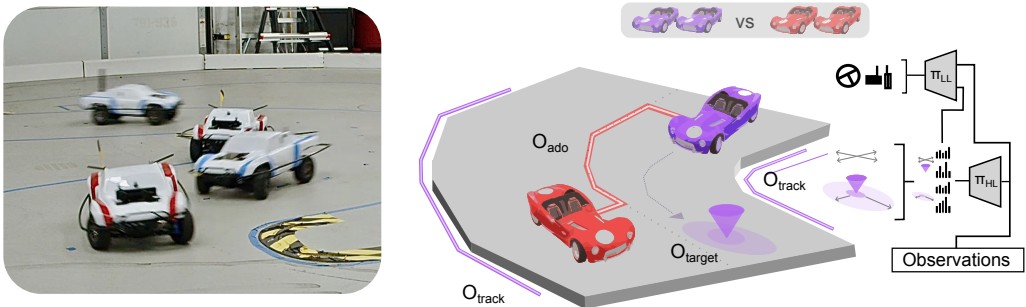

Figure 1: We train multi-team racing agents in simulation and transfer them directly to hardware. Our approach leverages a hierarchical policy as a layer of abstraction between strategic planning and continuous control. A high-level categorical policy predicts goal distributions for a low-level policy to achieve, effectively projecting long-term decision making into a limited preview horizon.

## 1 Introduction

Autonomous racing in a multi-vehicle team setting is a highly challenging task. It requires safely handling a vehicle at its dynamic limits while simultaneously reasoning about opponents and team strategy. These challenges push the limits of autonomous navigation and have, thus, attracted considerable academic interest. At the same time, solving autonomous racing promises to impact decision-making for real-world autonomous vehicles. The frequent interactions and driving at the limits of controllability are reminiscent of certain near-accident scenarios which are absent from common autonomous vehicle research datasets [1].

The multi-agent learning community has focused largely on simulated challenges such as [2, 3, 4]. We see multi-team racing as a step towards a simulation and hardware benchmark that is both

7th Conference on Robot Learning (CoRL 2023), Atlanta, USA.

accessible and realizable without the use of proprietary hardware, due to the vibrant open-source community revolving around scaled-car racing [5].

In the considered multi-team racing task, two teams of two agents compete for a team win. A team's rank is determined by its best-ranked team member at the end of a race. Solving this task effectively is challenging because it requires both long-term strategic planning (e.g. coordination with teammates or setting up overtaking maneuvers over multiple corners of the track) and fast, responsive short-term planning for swift and precise handling of the vehicle (e.g. to avoid collisions or to counteract unexpected slippage).

Most autonomous racing systems follow a highly structured pipeline similar to traditional driving pipelines. This allows for fine-grained design of each component within perception, planning, and control. In contrast, purely learning-based approaches to autonomous racing allow addressing fundamental questions of embodied AI research such as the emergence of social behavior, embodied learning setup, and safe real-world policy transfer. However, these have mostly been tested in simulated environments, due to the difficulty of large-scale experience data collection on real hardware. Simulations present many ways to diversify and extend data collection. In particular, simulating competitive multi-agent scenarios enables self-play with large agent populations and avoids potential damage to hardware systems. Yet to this day, sim-to-real transfer in domains where neither driving at the limits nor wheel-to-wheel contact dynamics are well modeled remains a challenge.

In this work, we demonstrate a sim-to-real approach to multi-team racing. We expand on successful hierarchical policy architectures and leverage parallelized self-play in simulation to learn competitive agents with a variety of emergent behaviors, that transfer onto real hardware. Toward this aim, our work makes the following contributions:

- A light-weight GPU-accelerated simulation environment for multi-team racing, augmenting first principles-based with data-driven models to facilitate highly parallelized learning.
- A hierarchical policy structure that serves as a layer of abstraction to disentangle long-term team-centric strategic planning from short-term ego-centric continuous control.
- Training in simulation with zero-shot transfer to scale race cars, demonstrating dynamic maneuvers in highly interactive wheel-to-wheel multi-team racing scenarios on hardware.

Specifically, we review related work in Section 2, followed by our approach for environment modeling and control in Sections 3 and 4, respectively. We describe details of our race car platform in Section 3.1, followed by results of transfer for wheel-to-wheel racing on hardware in Section 5.

## 2    Related Work

This work presents a platform that advances motion planning for autonomous racing, multi-agent reinforcement learning (MARL) applications, and practical approaches for dynamics modeling and sim-to-real transfer. Pushing the limits of autonomous vehicles through autonomous racing has seen a recent surge of interest [6]. Investigated systems include both reduced scale platforms [5, 7, 8], full scale platforms [9], as well as arcade-style simulated systems [10, 11, 12].

**Motion Planning**    Motion planners can be grouped into three main categories. First, obstacle avoidance planners treat other agents (henceforth referred to as *ado-agents*, [13]) as moving obstacles without explicit consideration of interactions between agents and their objectives. This includes collision-free path identification via dynamic programming [14], filter-based prediction [15], reinforcement learning (RL) [16], graph-based planning [17], and adapting a low-level optimization either via learned reward specifications [18] or trajectory warm-starts [19]. Second, game-theoretic planners explicitly represent interaction strategies of ado agents and iterate until an equilibrium is reached. This includes formulations as sequential bi-matrix games [20] or iterative dynamic programming in belief space [21], with analysis commonly limited to 1-vs-1 scenarios. Iterative best response for 6 vehicles is showcased in [22], with the prevalent drawback that game theoretic planners introduce a trade-off between solution quality and real-time capabilities. Finally, learning-based

planners only implicitly reason about interactions from prior data and do not compute explicit ado plans. In [23], hand-crafted scenarios and rewards are combined with model-free RL to beat human players in the game of Grand Turismo, while [11] learns a latent world-model that reasons about opponent behavior for competitive 1-vs-1 racing via self-play. The hierarchical approach in [12] enables a high-level policy to propose lane change sequences to a low-level controller in response to ado behavior. Our approach leverages a hierarchical policy structure as a layer of abstraction between team-centric strategic planning and ego-centric continuous control, learning efficient multi-agent interactions from experience and displaying emergent behavior via self-play. This further enables real-time inference when scaling to many-vs-many races on hardware.

**Multi-Agent Reinforcement Learning**   This work considers a mixed, cooperative-competitive, racing scenario, where teams receive a joint reward for their overall performance, with only very limited regularization terms signaling individual contribution. Similar problems are extensively studied in the MARL literature [24, 25, 26, 27]. In particular, we consider the centralized training decentralized execution (CTDE) paradigm with factored (action) value functions as described in [28, 29, 30]. These approaches attempt to learn individual payoff assignments between cooperating agents. This enables decentralized execution by greedy action with respect to the individual payoff. Alternatively factored policies can be directly trained using actor-critic approaches such as Proximal Policy Optimization [31]. We make use of CTDE to train the high-level policy with DecSARSA, an on-policy adaptation of decoupled Q-Networks [30], learning credit assignment for team members.

**Dynamics Modeling and Sim-to-Real Transfer**   To accelerate learning, we leverage massively parallel simulation and training running directly on the GPU [32]. A wide variety of dynamics models are used in practice for autonomous vehicles. For example, the authors in [33] use a dynamic single-track bicycle model with Pacejka tire models [34] and model randomization for policy learning. The same model formulation was used in [35] containing system identification of parameters for the F1Tenth platform. In [36], physics-informed basis functions are combined with Bayesian regression to fit a dynamics model used for predictive control. For a comprehensive overview of dynamics models for racing, see [6]. Finally, our work uses simulation-to-reality (sim-to-real) transfer. See [37, 38] for select surveys on sim-to-real approaches and [27] for MARL transfer onto hardware.

## 3   Simulation and Dynamics Modeling

Our approach leverages a highly parallelized, light-weight simulator for time-efficient behavior learning on GPUs. The simulator merges data-driven dynamics obtained through system identification on hardware with simplified analytic models to generate high-fidelity interactive scenarios.

### 3.1   Hardware Platform

We employ the TRIKart hardware platform developed by the Toyota Research Institute [8], a variation of the F1Tenth vehicle design [5]. Each vehicle has an Nvidia Jetson onboard computer to handle communication with the VESC motor driver, steering servo, and the base station computer. The base station provides real-time control commands together with state estimates obtained from an OptiTrack motion capture system. We further outfit each vehicle with a Lexan cover augmented with ABS bars to mitigate damage during vehicle-to-vehicle contact scenarios.

### 3.2   Dynamics Model

The simulator leverages a switched dynamics model conditioned on the contact state of the system. Out of collision, we employ the kinematic bicycle model [39] augmented with data-driven models. During collisions, we employ the dynamic bicycle model [14]. Both variations are implemented as elementary tensor operations in PyTorch for rapid data generation [40], and described below.

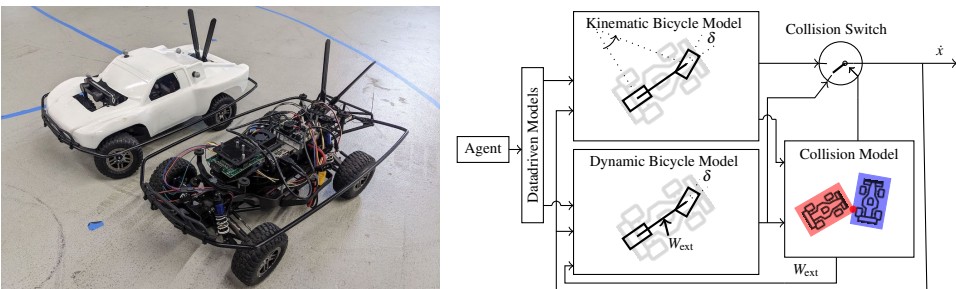

Figure 2: *Left* – 10th-scale TRIKart platform. *Right* – Block diagram of the switched dynamics. The simulator augments a kinematic bicycle model with data driven front slip angle and drive train dynamics learned from the real vehicle. If a collision event is detected, the dynamics are switched temporarily to a lower fidelity dynamic model that directly models the effects of external wrenches.

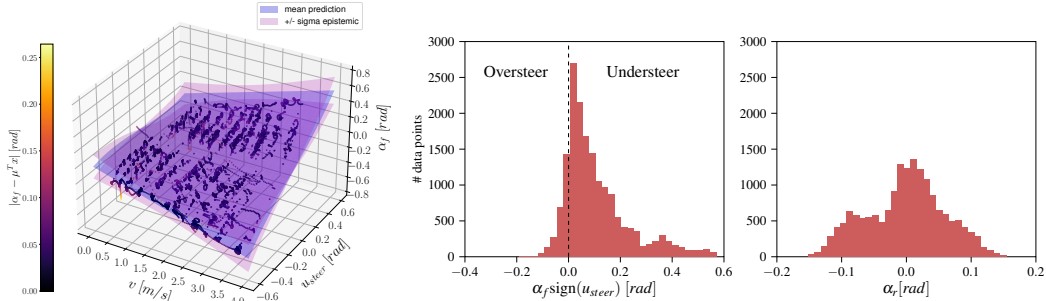

Figure 3: A linear model with odd steering angle features is fit on hardware data using BLR to capture model mismatches based on understeer. *Left* – Model predictions with epistemic uncertainty used for automatic domain randomization. *Right* – Histograms of front, $\alpha_f$, and rear slip angle, $\alpha_r$. The front slip angle is scaled such that positive values correspond to understeer at the front axle, which covers around 87% of the recorded data.

**Out-of-contact model:** We combine the kinematic bicycle model from [39] with feature-based predictors of the front tire slip angle as well as a data-driven model of the longitudinal dynamics (see Section 3.3). The dynamics of the kinematic model are then given by

$$\dot{x} = f_{\text{kin}}(x, u) = \begin{bmatrix} v\cos(\theta + \beta) \\ v\sin(\theta + \beta) \\ \frac{v\tan(u_{\text{steer}} + \alpha_f)}{\sqrt{L^2 + (l_r\tan(u_{\text{steer}} + \alpha_f))^2}} \\ \Phi_{\text{long}}(v, u_{\text{steer}}, u_{\text{acc}}), \end{bmatrix}, \tag{1}$$

where $x = [X, Y, \theta, v]$ is the vehicle state, $u = [u_{\text{steer}}, u_{\text{acc}}]$ are the controls, the center of mass slip angle is $\beta = \arctan\left(\tan(u_{\text{steer}} + \alpha_f)\frac{l_r}{l_r + l_f}\right)$, $\alpha_f$ is the predicted front slip angle, and $\Phi_{\text{long}}$ represents the longitudinal acceleration model. The data-driven augmentation yields a sufficiently high-fidelity representation of the hardware dynamics, but does not explicitly model reaction forces. We identify collisions based on rectangular hit boxes and simulate contact wrenches with a spring model accounting for penetration depth.

**In-contact model:** When a collision event is detected, the simulation switches to the dynamic bicycle model from [14, 35]. The resulting contact wrenches are then explicitly propagated into the state dynamics, switching back to the kinematic model after a pre-specified cool-down or if the rear slip angle falls below a threshold. A schematic of the switched dynamics is provided in Figure 2.

### 3.3 System Identification

The parameter identification and data-driven model estimation proceed directly on the hardware system. The steering commands are calibrated based on trajectories recorded with a motion capture

system under the condition of minimal tire slip, allowing for the steering angle to be inferred from the center of mass slip angle. While the resulting kinematic bicycle model provides a good initial approximation of the dynamics, it does not capture higher-order effects relating to e.g., variable drive-train acceleration or tire traction. To better model these effects, we collected a dataset of trajectories exciting the full range of throttle and steering inputs, while avoiding the oversteer regime due to the vehicles' tendency to understeer when running at the dynamic limits on our track. See Figure 3. A longitudinal acceleration model is then trained to minimize the one-step prediction error based on velocity, steering, and throttle inputs. To model lateral deviations at higher speeds we estimate the slip angle of the front tire via Bayesian linear regression (BLR). Hereby we use simple hand crafted features that are odd in the steering angle to fit our data. We assume our measured slip stems from a linear model in the features with i.i.d. Gaussian measurement noise $\varepsilon \sim \mathcal{N}(0, \sigma_n^2)$ and a spherical Gaussian prior on the model weights $\varepsilon_w \sim \mathcal{N}(0, \sigma_p^2 \mathbb{I})$ . During training the dynamics are randomized by drawing slip models from the posterior distribution and injecting noise into the received commands. See Appendix A and C.4 for details.

# 4 Agent Structure and Training Process

We define multi-team racing as a finite-horizon decentralized MDP (decMDP) [41, 42] (brief review in Appendix D). We leverage a GPU-accelerated implementation of the dynamics described in the previous section to train competitive multi-team racing agents. Our approach combines coarse high-level planning with fine-grained low-level control within a hierarchical policy structure trained via bilevel optimization. An overview of the bilevel learning procedure is illustrated in Figure 4.

## 4.1 Hierarchical Control Structure

Competitive control in interactive settings requires both long-term strategic decision-making as well as short-term responsive action selection. These objectives occur at different layers of abstraction, with low-level control conditioned on high-level decisions, thus lending themselves to bilevel optimization procedures [43, 44, 45, 46]. Here, we leverage a hierarchical policy design [47, 48] to effectively guide ego-centric low-level decision-making through team-centric high-level goals. To this end, we train a discrete high-level policy to suggest goal states, and a low-level policy to resolve local velocity and heading commands in order to achieve these goals, as illustrated in Figure 1.

**High-level control:** Our formulation enables the high-level policy to propose goal states to the low-level controller. This introduces a layer of abstraction that helps to disentangle long-term strategic planning from short-term control. Goal states are encoded by their relative position along the track as well as their associated uncertainty. The uncertainty encodes goal importance with respect to the high-level strategy and enables the low-level policy to weight short-term gains against long-term considerations. We further leverage coarse high-level representations via categorical distributions to encode mode-switching behavior, e.g., *"slow down and keep to the right"*. This encourages diverse high-level strategies and can provide favorable exploration characteristics [49]. The high-level policy is obtained as an $\varepsilon$-greedy evaluation of a state action value function trained with Decoupled SARSA, an on-policy variation of Decoupled Q-Networks [30]. This leverages value decomposition across team members and enables Centralized Training with Decentralized Execution (CTDE) [28, 50]. We further adopt Munchausen Value Iteration [51] for regularization.

**Low-level control:** The low-level controller receives high-level goals with associated time budgets, predicting steering and velocity commands. This effectively augments short-horizon local reasoning with long-horizon strategic planning. However, the goal states only act as soft constraints, enabling the low-level controller to retain flexibility and prioritize local decision-making when necessary. The low-level policy is trained via PPO [52], with temporal difference (TD) learning of the values and Generalized Advantage Estimation (GAE) of the policy, under the primary objective of clearing high-level goals, while considering behavior smoothness, obstacle avoidance, and local track geometry. The strategic team decisions are thereby implicitly provided through high-level goals.

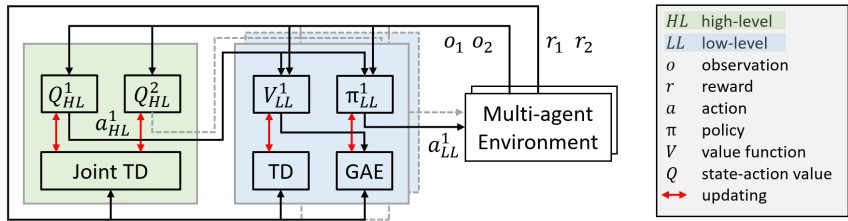

Figure 4: The hierarchical policy structure abstracts team-centric strategy from ego-centric control. The high-level predicts discrete goal distributions and is trained jointly across team members leveraging value decomposition with DecSARSA. The low-level learns continuous actions to satisfy individual goal states with PPO. Training proceeds in parallel simulation via bilevel optimization.

**Hardware-level control:** The steering and velocity commands sampled from the low-level policy serve as references to a hardware-level PD tracking controller. The PD controller generates steering and acceleration commands, running at a faster rate to mitigate the effects of unmodeled dynamics.

## 4.2 Training in Simulation

The hierarchical policy is trained via bilevel optimization in parallel simulation. Training consists of two phases, learning basic driving skills during pre-training and effective racing during fine-tuning.

**Pre-training:** Initially, the low-level policy will have to learn rudimentary driving skills that enable goal-reaching behavior. During this stage, the agent will only generate low quality learning signals for the high-level policy that may induce representational collapse [53]. We therefore pre-train the low-level policy for 500 iterations with PPO on goals sampled from the initially uniform high-level policy, competing against Pure Pursuit opponents to simulate capable ado behavior.

**Fine-tuning:** The bilevel optimization then alternates every 50 iterations. High-level training leverages DecSARSA to refine strategic behavior via goal prediction, while low-level training employs PPO to align local controls with the adapted high-level goal distribution. During this stage, we sample opponents as a mixture from prior policy checkpoints to encourage competitive behavior to emerge through self-play, and opponents steered by pure-pursuit control to prevent strategic collapse [54]. We consider both joint and independent high-level Q-functions for fine-tuning, referring to the two approaches as RL Team and RL Independent.

**Observations and rewards:** The high-level policy observes ego state, rank information, and track boundaries over a preview horizon. It further receives noisy ado odometry data within a view-limited neighborhood, which is processed by a multi-head attention encoder to extract ado relevance with respect to the current ego state into a latent representation [55]. The low-level observations additionally contain the high-level goals, as in Figure 1. The high-level policy is primarily rewarded based on team rank information, encouraging strategic behavior that improves positioning of the ego team, and receives zero-sum overtaking reward and a small term encouraging ego progress. Hereby, the dominating term is chosen to be a sparse end-of-episode team-rank reward. The low-level policy is primarily rewarded for accomplishing high-level goals at their respective timings, with regularization objectives that modulate vehicle collisions, off-track driving, and control smoothness. Further details regarding the observation and reward structure are provided in Appendices B and C.

## 5 Experiments

We evaluate the efficacy of our approach by first training policies in simulation and then transferring them directly to the scale race car hardware. Policy learning is conducted on a single Nvidia RTX 2080Ti with 2048 parallel environments taking roughly 8 hours. Policy inference is conducted on the CPU for all experiments with the high-level policy running at 0.5 Hz and the low-level policy running at 20 Hz. The employed learning parameters and training details are summarized in Appendix C.

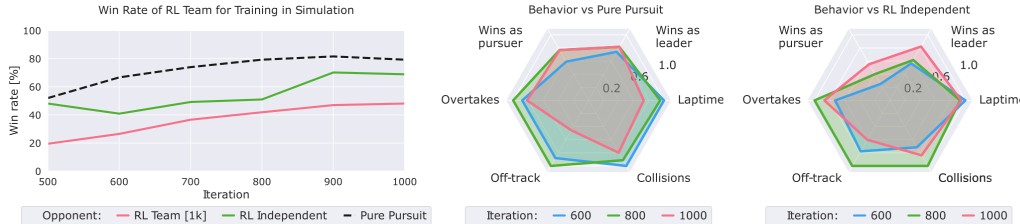

Figure 5: Evaluation in simulation shows consistent improvement of winrate and driving skill over the course of training. We compare our RL Team agent to an egoistic variation with independent high-level learners and a pure pursuit baseline, achieving a winrate of 79% and 69%, respectively. The behavior profiles indicate how increases in winrate yield complex trade-offs between lower lap-times, risk-seeking, and risk-averse behavior (e.g. blocking increases laptime and may risk crashes).

The results from fine-tuning in simulation are summarized in Figure 5. We evaluate winrates of the RL Team agent over the course of training against itself at convergence, a variation that uses independent learning, and the pure pursuit baseline. We further quantify behavior characteristics based on races against the latter two agents. The winrate of RL Team steadily improves to reach around 80% against pure pursuit and 70% against RL Independent. In particular, winrate from behind approaches 60% against RL Independent, while laptime and time spent off-track are reduced. The interplay of other factors is more complex, as e.g. emergence of blocking will initially induce larger deviation from the racing line in high-risk maneuvers that other agents may exploit, while skillful blocking will reduce the likelihood of crashes and the need to re-overtake later on. Generally, we find high-level training to significantly improve performance, yield better positioning on the track, and enable smoother driving.

We further evaluate our approach against two model predictive path integral control (MPPI) [56] agents, running collision avoidance based on an MPPI motion predictions for all 3 ado cars (see [57]), and against two RL Independent agents on hardware for 20 five-lap races each. The 20 races are split into 3 starting configurations: *Front* – RL Team starting in 1st and 2nd (for 5 races), *Back* – RL Team starting in 3rd and 4th place (for 5 races), and *Random* – randomized starting configurations (for 10 races). The maximum velocity is set to 3.5 [$\frac{m}{s}$] for all hardware experiments. The results from the hardware evaluation are summarized in Figure 6. On hardware, we observed single-agent average laptimes across 10 laps of 6.8s for RL Team, 7.6s for RL Independent and 6.7s for MPPI. We found that RL Team was great at keeping its position once in front, for example by using the rear team member to fend off opponents, resulting in a 92% winrate against MPPI and a 100% winrate against RL Independent, when starting in 1st place aggregated over *Front* and *Random* scenarios. We further observe that RL Independent agents display a more reckless driving style reflected in the number of collisions, crashes, and offtrack time, see Figure 6, which forces the RL Team agents to negotiate an increased number of (near-)collision events as reflected by the behavioral differences in the radar charts. We refer the reader to Appendix E for the un-normailzed data.

Emergent behavior observed on hardware is showcased in Figure 7. The hierarchical policies learn multi-car overtaking maneuvers, altruistic blocking where the rear team-member significantly slows down and attempts to block attacking opponents, and pitting maneuvers to jostle opponents out of the way, yielding successful sim-to-real transfer of competitive behaviors. The emergent teaming be-havior is facilitated by the centralized high-level controller that sets distinct goals for and distributes team success to individual members, allowing for altruistic agents that sacrifice own performance by blocking opponents to profit from the resulting increase in win-chance of their team member (Figure 7, middle). We refer the reader to Appendix F for further simulated examples.

## 6 Limitations

We briefly discuss limitations to our current approach, which point towards avenues for future work. **Ado observation sensitivity**: Agents retain sensitivity to their ado observations even when strategic interactions are not likely. This was more pronounced during hardware experiments when coupled

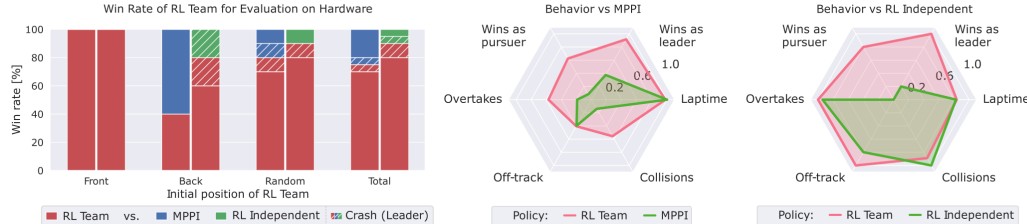

Figure 6: Transfer to hardware shows strong performance of our RL Team agent compared to the RL Independent and MPPI baselines, achieving winrates above 80% and 70%, respectively. We observe consistent wins when starting in front and a strong ability to win from the back of the starting grid. Stripes indicate crash scenarios in which a pursuing agent did not finish the race, colored by winner.

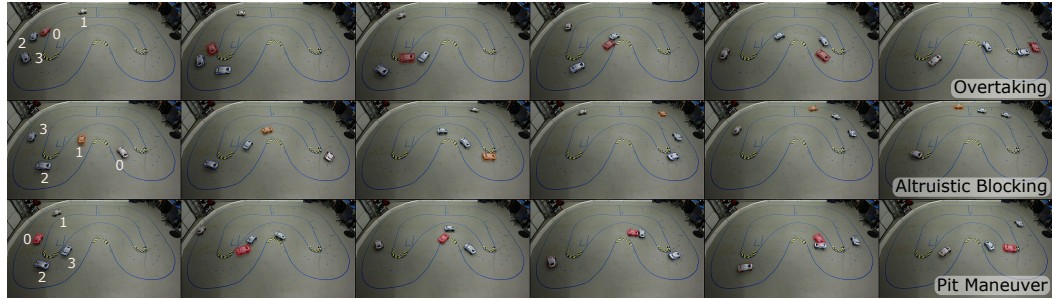

Figure 7: Three observed emergent behaviors. *Overtaking*: Agent 0 performs a double overtake on two ado agents. *Altrusitic Blocking*: Agent 1 stays back to block agents 2 and 3, allowing for agent 0 to pull away. *Pit Maneuver*: Agent 1 pulls up behind agent 3 and performs a pitting maneuver. See also supplementary material for video sequences.

with system delays and the sim-to-real gap of the dynamics model, sometimes causing exaggerated deviations from the race line. **Collisions between teammates and corner cutting**: Teammates occasionally jostle for positions, and trade penalties for potential position gains by cutting corners, which could be refined by adapting the problem formulation. **Dynamic limits**: While this work brings the vehicles to the limits of friction and demands corrections for understeer, the presented pipeline does not allow for agents to act at the torque limits of the scaled cars. Future work could investigate this dynamics regime while exploring methods to further reduce hardware system delays.

## 7 Conclusion

We propose a novel light-weight simulator for learning dynamic multi-team wheel-to-wheel racing for zero-shot sim-to-real transfer. The simulator combines classical analytic models with data-driven components and is GPU-accelerated to enable massively parallel simulation. We further introduce a hierarchical control structure that disentangles high-level strategic planning from low-level continuous control. The high-level policy suggests goals to the low-level controller and is represented with categorical variables to encode mode-switching behavior. The low-level policy predicts continuous controls and incorporates high-level guidance into its local decision making, but retains the ability to deviate if required. The high-level policy is obtained from centralized training across team members and therefore provides strategic information beyond single-agent control.

We train multi-agent racing policies in simulation and deploy onto the TRIKart scale race car platform without further adaptation. The transferred policies retain their efficacy and are able generate dynamic behavior on hardware when tested with a team of MPPI ado agents, including overtaking, blocking, and even pit maneuvers. Future work will be directed at further reducing the sim-to-real gap, while tackling hardware system engineering challenges in supporting more ado agents. Generally, our results indicate great promise for combining parallelized simulation with hierarchical policy abstraction for achieving zero-shot sim-to-real transfer of dynamic multi-team racing behaviors.

**Acknowledgments**

This work was supported in part by Toyota Research Institute (TRI). This article solely reflects the opinions and conclusions of its authors and not TRI, Toyota, or any other entity. We thank them for their support. We further thank Velin Dimitrov for assistance with hardware deployment and Markus Wulfmeier for fruitful discussions on DecSARSA.

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

# A  System Identification Details

In order to identify the slip model we use Bayesian Linear regression. Analogously to [58], Chapter 3, we compute the posterior and inference distributions at $\hat{x}$ as

$$p(w_{\text{slip}}|X,Y) = \mathcal{N}(w_{\text{slip}}; \mu, \Sigma) \tag{2}$$

$$\mu = \left(X^T X + \frac{\sigma_n^2}{\sigma_p^2}\mathbb{I}\right)^{-1} X^T Y \tag{3}$$

$$\Sigma = \left(\frac{1}{\sigma_n^2}X^T X + \frac{1}{\sigma_p^2}\mathbb{I}\right)^{-1} \tag{4}$$

$$p(\hat{y}|\hat{x},X,Y) = \mathcal{N}(\hat{y}; \mu^T\hat{x}, \underbrace{\hat{x}^T\Sigma\hat{x}}_{\text{epistemic}} + \underbrace{\sigma_n^2}_{\text{aleatoric}}) \tag{5}$$

where the resulting uncertainty splits into an epistemic and an aleatoric component. In order to fit the model we use the following features and targets,

$$X = \begin{bmatrix} | & | & | \\ u_{\text{steer}} & v u_{\text{steer}} & u_{\text{steer}}^3 \\ | & | & | \end{bmatrix} \quad Y = \begin{bmatrix} | \\ \alpha_f \\ | \end{bmatrix} \tag{6}$$

along with hand-tuned noise parameters in Table 1.

| $\sigma_p^2$ [rad] | $\sigma_n^2$ [rad] |
|---|---|
| 1.0 | 0.01 |

Table 1: Employed BLR noise parameters.

# B  Observation Details

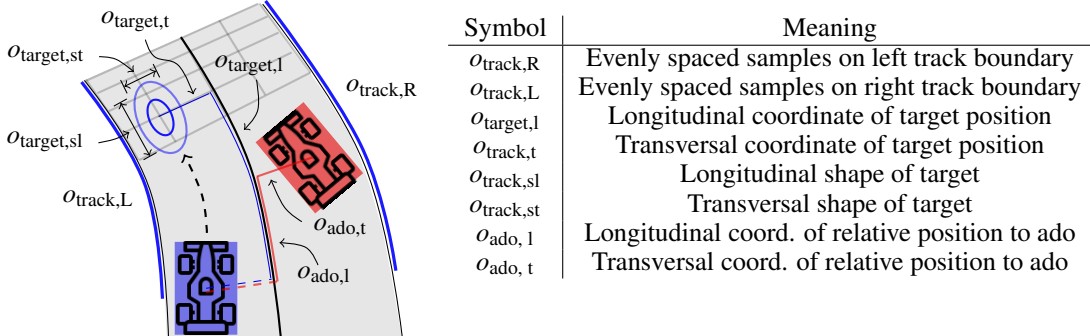

| Symbol | Meaning |
|---|---|
| $o_{\text{track,R}}$ | Evenly spaced samples on left track boundary |
| $o_{\text{track,L}}$ | Evenly spaced samples on right track boundary |
| $o_{\text{target,l}}$ | Longitudinal coordinate of target position |
| $o_{\text{track,t}}$ | Transversal coordinate of target position |
| $o_{\text{track,sl}}$ | Longitudinal shape of target |
| $o_{\text{track,st}}$ | Transversal shape of target |
| $o_{\text{ado, l}}$ | Longitudinal coord. of relative position to ado |
| $o_{\text{ado, t}}$ | Transversal coord. of relative position to ado |

Figure 8: Track and position-based ado observation components employed to train the hierarchical policies.

The high-level observations consist of the ego observations and ado observations

$$o_{\text{HL}} = \begin{pmatrix} o_{\text{ego}} \\ o_{\text{ado}} \end{pmatrix} \tag{7}$$

where $o_{\text{ado}}$ is the concatenation of all individual, view-limited, ado observations $o_{\text{ado, i}}$. The individual components read

$$o_{\text{ego}} = \begin{pmatrix} v \\ \dot{\theta} \\ o_{\text{track,R}} \\ o_{\text{track,L}} \\ t_{\text{episode}} \\ o_{\text{wot}} \\ o_{\text{progress}} \\ \lambda^E \\ \lambda^T \end{pmatrix}, \qquad o_{\text{ado, i}} = \begin{pmatrix} o_{\text{ado,l}} \\ o_{\text{ado,t}} \\ \sin(\theta_{\text{rel}}) \\ \cos(\theta_{\text{rel}}) \\ v_{\text{rel}} \\ \dot{\theta}_{\text{rel}} \end{pmatrix}, \tag{8}$$

where $t_{\text{episode}}$ is the fraction of the high-level episode remaining, $o_{\text{wot}}$ is the fraction of wheels off track, $o_{\text{progress}}$ is the amount of progress along the center line of the track over the last time step, $\lambda^E$ and $\lambda^T$ are the ego and team rank respectively.

The low-level observations are the concatenation of the high-level observations with the target position provided by the high-level policy, where distances are expressed in track coordinates, such that

$$o_{\text{LL}} = \begin{pmatrix} o_{\text{HL}} \\ o_{\text{target, l}} \\ o_{\text{target, t}} \\ o_{\text{target, sl}} \\ o_{\text{target, st}} \end{pmatrix}. \tag{9}$$

## C    Learning Parameters and Domain Randomization

We elaborate on aspects of the training setup below, discussing the reward functions and domain randomization used. Throughout, we consider high-level steps that consist of $T_{LL} = 20$ low-level steps. At each high-level step, a goal state is sampled and then held fixed for the duration of the low-level substeps.

### C.1    PPO hyperparameters

In order to train the low-level controller we leverage Proximal Policy Optimization (PPO) with Generalized Advantage Estimation (GAE) similar to the parallelized agent employed in [32]. The associated hyperparameters are provided in Table 2. The low-level controller generates single agent driving behavior based on local observations, while receiving guidance from the high-level policy through goal-conditioning.

| Parameter | Value |
|---|---|
| Learning rate | $1 \times 10^{-4}$ |
| Minibatch size | 512 |
| Minibatches | 4 |
| Learning epochs | 2 |
| Clip value | 0.2 |
| Discount ($\gamma$) | 0.99 |
| GAE parameter ($\lambda$) | 0.95 |
| Entropy coefficienct | 0.01 |
| Desired KL | 0.01 |
| Max grad norm | 10.0 |
| MVI $\alpha$ | 0.9 |
| MVI $\tau$ | 0.03 |

Table 2: Algorithm hyperparameters

## C.2 Decoupled Expected SARSA

In order to train the high-level controller we build on recent results at the intersection of parallel optimization and multi-agent control and employ Decoupled Expected SARSA, an on-policy variation of Decoupled Q-learning [30] within the family of Hypergraph Q-Network algorithms [59]. This approach leverages value decomposition [28] across action dimensions and team members to represent the state-action value function as

$$Q_\theta(\boldsymbol{s}_t, \boldsymbol{a}_t) = \sum_{j=1}^{M} \frac{Q_\theta^j(\boldsymbol{s}_t, a_t^j)}{M}, \tag{10}$$

where coordination is facilitated by individual utility functions being conditioned on the global robot state and a high-degree of parameter sharing within a unified critic [60]. The linear combination of univariate utility functions allows for efficient decomposition of the argmax operator and global optimization over $\boldsymbol{a}_t$ simplifies into parallel local optimizations over $a_t^j$. Training of the action value function remains centralized, while online action selection is decoupled. Inserting this decomposition into the Bellman equation yields a decoupled target representation with expectation over the underlying policy (e.g. $\varepsilon$-greedy)

$$y_t = r(\boldsymbol{s}_t, \boldsymbol{a}_t) + \gamma \sum_{j=1}^{M} \pi(\boldsymbol{s}_{t+1}, a^j) \frac{Q_\theta^j(\boldsymbol{s}_{t+1}, a_{t+1}^j)}{M}. \tag{11}$$

We can then insert the factorized value function of Eq. 10 and the decoupled target of Eq. 11 into the loss function $\mathscr{L}(\theta) = \sum_{b=1}^{B} L_\delta(y_t - Q_\theta(s_t, a_t))$. We furthermore explore the effects of soft Q-learning [61] within the context of Munchausen Value Iteration [51] to yield the adapted target

$$y_{t,MVI} = r(\boldsymbol{s}_t, \boldsymbol{a}_t) + \alpha\tau\ln\pi_\theta(a_t^j|\boldsymbol{s}_t) + \gamma \sum_{j=1}^{M} \pi(\boldsymbol{s}_{t+1}, a^j) \frac{Q_\theta^j(\boldsymbol{s}_{t+1}, a^j) - \tau\ln\pi_\theta(a_{t+1}^j|\boldsymbol{s}_{t+1})}{M}, \tag{12}$$

where $\pi_\theta = \text{sm}(\frac{Q_\theta}{\tau})$ is now taken as the softmax policy with temperature $\tau$ and scaling factor $\alpha$. We observed improved training using Munchausen Value Iteration and trained our final policies based on this formulation. The associated hyperparameters are provided in Table 2.

## C.3 Reward functions

We provide details on the reward function used to train the high-level (HL) and low-level (LL) policies. For ease of notation, we introduce the following variables: $\lambda^T$ and $\lambda^E$ represent team and ego rank, respectively, $p$ and $\dot{p}$ denote the vehicle position and velocity in track frame, respectively, $\mu^g$ and $\Sigma^g$ correspond to the mean and standard deviation of the multivariate goal state distributions, respectively, $w_i$ is an indicator for whether tire $i$ is off-track, $a$ represents the actions, and $\xi$ is a collision indicator. The wheel-on-track term features a slight asymmetry that up-weights penalty terms when either the front or rear axle are off-track, while the collision term up-weights rear-ending collisions - we omitted these features from the reward equations for visual clarity. We further introduce index $t$ denoting time and $t_{LL}$ as the fraction of time left until the next high-level step.

$$r_t^{HL} = c_1(\lambda_t^T == 0) + c_2(\lambda_0^T - \lambda_t^T) + c_3 \frac{\lambda_{t-1}^T - \lambda_t^T}{1 + \min(\lambda_{t-1}^T - \lambda_t^T)} + c_4 e^{-\lambda_t^T} + c_5 e^{-\lambda_t^E} \tag{13}$$

$$r_t^{LL} = c_6 \frac{1}{2\pi|\Sigma_t^g|^{1/2}} e^{-\frac{1}{2}(p_t-\mu_t^g)^T {\Sigma_t^g}^{-1}(p_t-\mu_t^g)+\frac{1}{4}\dot{p}_t} + c_7 \sum_{i=0}^{3} w_t^i + c_8\Delta p_t + c_9 e^{-|\dot{p}|} + c_{10}(a_t - a_{t-1})^2 + c_{11}\xi_t \tag{14}$$

To provide intuition for the effect of individual reward terms, we discuss the high-level reward function in more detail. The high-level episode duration is set to 16 seconds and low-level episodes terminate after two high-level targets have been received. The first place reward ($c_1$) provides a sparse signal about team performance to each team member at each timestep and serves as the primary indicator for winning a race. The starting rank reward ($c_2$) acts as a small regularizer that

| | Symbol | Value | Term explanation |
|---|---|---|---|
| | $c_1$ | 4.0e-1 | Team is in first place |
| | $c_2$ | 5.0e-2 | Team improved over starting rank |
| HL | $c_3$ | 1.0e-1 | Zero-sum overtaking reward |
| | $c_4$ | 1.0e-3 | Regularize with team rank |
| | $c_5$ | 1.0e-2 | Regularize with ego rank |
| | $c_6$ | $\mathbb{I}(t_{LL} < 0.1) * (1.0e+0)$ | Clear goal at end of low-level episode |
| | $c_7$ | -1.0e-3 | Penalize wheels off track |
| LL | $c_8$ | $\mathbb{I}(\Delta p_t > 2.5) * (-2.5e-1)$ | Penalize large progress jumps (cutting) |
| | $c_9$ | -2e-3 | Encourage high velocity |
| | $c_{10}$ | -6e-4 | Regularize action smoothness |
| | $c_{11}$ | -5e-3 | Penalize collisions |

Table 3: Reward function components

up/down-weighs trajectories based on their team rank change, as winning from behind requires more skill than winning from the front. The overtaking reward ($c_3$) explicitly highlights important sparse interactions that resulted in team rank change to pinpoint the influence of particular short-term maneuvers on long-term race strategy. The team rank reward ($c_4$) complements the first place reward ($c_1$) by encouraging lower team rank even if the team is not in first place, providing dense guidance during the learning process. The ego rank reward ($c_5$) further encourages agents to improve their individual rank even if not impacting overall team rank. The latter term primarily act as regularizers that counteracts potential lazy agents to e.g. encourage a team member in last place to remain competitive even when their team is in first place and they do not foresee short-term individual impact. Note that given $c_1$ in Table 4, the sparse high-level reward outweighs the highest possible accumulated dense reward components, which are weighted by $c_4$ and $c_5$.

## C.4 Domain Randomization

To improve the sim-to-real transfer of the learned policies we randomize the dynamics simulation during training time. The simulated slip is randomized by sampling a new weight vector from the posterior distribution in Equation 2 at the beginning of every training episode. We further randomize select model parameters with additive noise $\delta$ summarized in Table 4.

| Symbol | Distribution | Explanation |
|---|---|---|
| $v_{\max}$ | $\mathscr{U}(3.5, 4.5)[\frac{m}{s}]$ | Additive noise velocity cap |
| $\delta_{l_f}$ | $\mathscr{U}(-0.02, 0.02)[m]$ | Additive noise to front length |
| $\delta_{l_r}$ | $\mathscr{U}(-0.02, 0.02)[m]$ | Additive noise to rear bicycle length |
| $\delta_{u_{\text{steer}}}$ | $\mathscr{U}(-0.03, 0.03)[rad]$ | Additive noise to all steering commands |
| $\delta_{u_{\text{acc}}}$ | $\mathscr{U}(-0.15, 0.05)[-]$ | Additive noise to simulated throttle commands |

Table 4: Ramdomized model parameters

In order to improve robustness of the policies we add uniform random noise, drawn independently from $\mathscr{U}(-0.005, 0.005)$, to all observations except the rank and team rank observations.

## D  Decentralized Markov Decision Process Formulation

We formulate multi-team racing as a finite horizon decentralized Markov Decision Process (DecMDP), that is characterised by the tuple $(\gamma, S, T, A, \Omega, O, R)$, where $\gamma$ is the discounting factor, S is the set of joint states, $T$ is the probability densitiy fucntion $p(s'|s, a)$ of transitioning from $s$ to $s'$, when agents take the joint action $a$, $A$ is the set of action sets available to all agents, $\Omega$ is the set of all observation spaces, $O$ is the set of all observation functions, and $R$ is the set of reward functions across agents, where each reward function returns the reward for taking joint action $a$ at state $s$.

In practice, we define multi-team racing in a symmetric fashion across agents such that their action spaces are identical. The observation functions are defined as outlined in Section 4, such that each agent only observes their velocity, nearby track boundaries, number of wheels that are off track, rank, remaining time in the episode, last actions, and respective ado agents.

# E    Hardware Experiments

The radar plots in Figure 6 are intended to give the reader an intuitive understanding of the behavior of the various agents. For clarity the axis are normalized with the maximum value of the respective statistic aggregated during both the races against of RL Team (RL T.) against MPPI agents and RL Independent (RL Ind.). In this appendix the raw data is listed in Table 5, along with brief explanations. All statistics are averaged over the 20 respective races. As a guideline, the average race duration is around 40s.

| Statistic | RL T. vs MPPI | RL T. vs RL Ind. | Explanation |
|---|---|---|---|
| Laptime [s] | (8.44 , 8.62) | (8.36, 8.26) | Avg. Laptime of fastest team member |
| Wins as leader | (0.92, 0.38) | (1.00, 0.2) | Win rate w/ team member starting in $1^{st}$ |
| Wins as pursuer | (0.63, 0.08) | (0.80, 0.00) | Win rate w/ opponent starting in $1^{st}$ |
| Overtakes | (2.85, 0.95) | (5.00, 4.70) | Number of overtakes per team |
| Off-track [s] | (1.57, 1.54) | (3.89, 3.104) | Time off track per team |
| Collisions | (1.0 , 0.25) | (1.6, 1.8) | number of collisions per team member |

Table 5: Un-normalized data from the radar charts in Figure 6.

## F  Emergent Behavior in Simulation

We observe similar emergent behavior in simulation as on hardware. This is shown in Figure 9. The simulator itself features a viewer with visualizations of the command distributions of the high and low-level policies, markers for the observations, and miscellaneous additional information, see Figure 10.

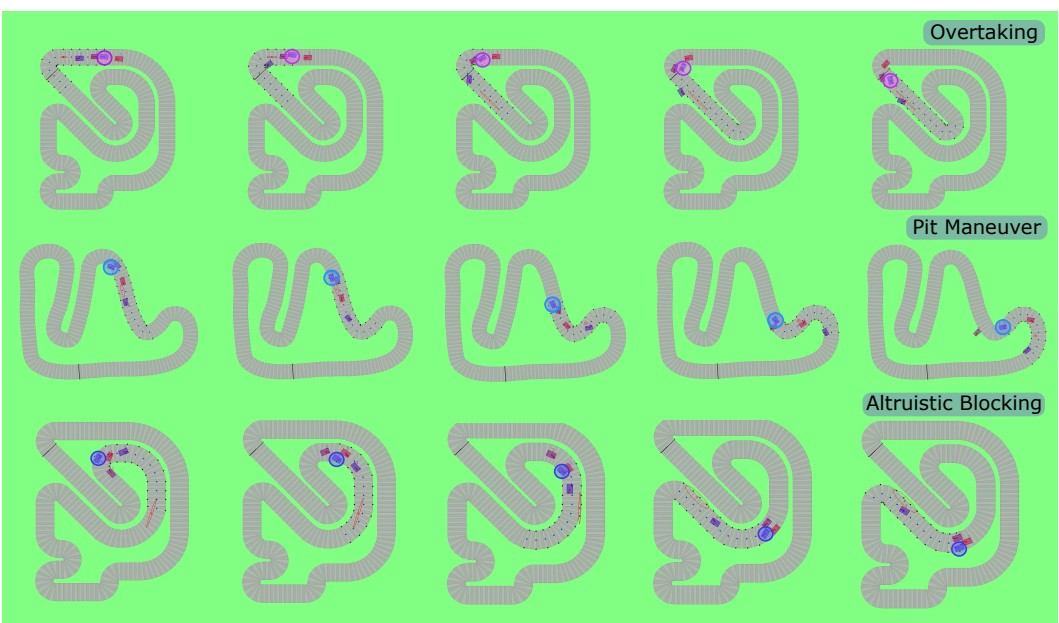

Figure 9: Three observed emergent behaviors in simulation. Overtaking: The marked agent performs an overtake. Pit Maneuver: The marked agent pulls up behind the red agent and performs a pitting maneuver. Altrusitic Blocking: The marked agent overtakes an opponent and stays back to block both opponents allowing the blue teammate to pull away.

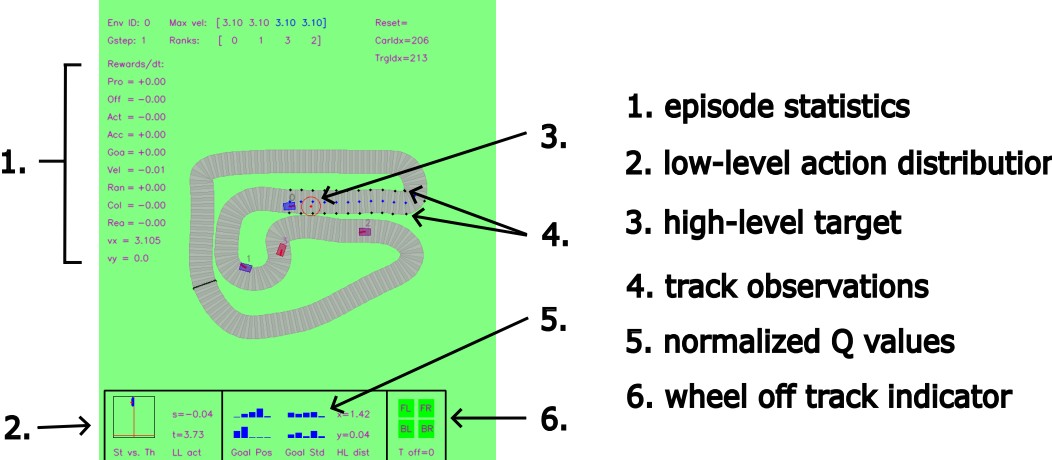

Figure 10: The viewer of the simulator displays individual environments along with episode information and information about the distributions of both the high and the low-level policy.

# G   Tracks used for Training

In Figure 11 the tracks employed for training and evaluation are shown. During training and evaluation, races take place both in the clockwise and counterclockwise direction. Track b) is a re-scaled version of the track used in [14].

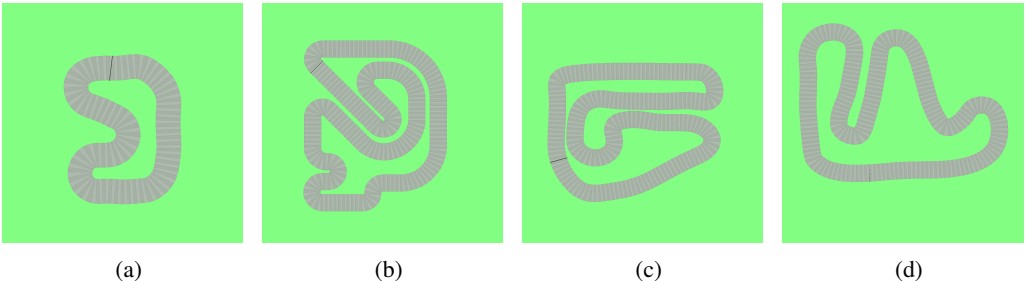

(a)                    (b)                    (c)                    (d)

Figure 11: Tracks used for training hierarchical racing policies. Track a) is used for the hardware evaluation.

