# OpenReview forum: "Dynamic Multi-Team Racing: Competitive Driving on 1/10-th Scale Vehicles via Learning in Simulation"
_robot-learning.org/CoRL/2023/Conference — CoRL 2023 Poster_

### Official Review · Reviewer_x7zC · 2023-07-17

**Confidence:** 5
**Originality:** Good
**Technical Quality:** Very Good
**Clarity Of Presentation:** Excellent
**Impact:** 3

**Recommendation:**

Strong Accept: I recommend accepting the paper and will argue for my recommendation even if other reviewers hold a different opinion.

**Review:**

Quality of the Research Paper

Originality:
- The originality of the paper lies in the difficulty of the high-speed/ high-acceleration and proximity overtaking maneuvers. Racing itself is a difficult task and many authors try to created modular software pipelines in order to create autonomous racing algorithm. With the self-play reinforcement learning approach it was shown that multi-agent competitive driving and overtaking can be demonstrated.
- By focusing on teams of cars, the chance of multi-team learning is given. This is a aspect that could help to transform DRL further to real-world autonomous cars e.g. in overtaking or merging scenarios

Significance of the work:
- Although autonomous racing is a niche field,  the study addresses an important and relevant research problem and therefore making a valuable contribution to the existing autonomous racing community. Especially the task of multi-vehicle overtaking was rather underrepresented so far.

Strength of the work:
- the paper addresses an important and timely research question and contributes to the existing body of knowledge in the field of field.
- Well-organized, structured and cited state of the art
- Clear and Coherent Structure: The paper exhibits a well-organized structure, making it easy for readers to follow the flow of ideas and understand the research process.
- Comprehensive Approach: The paper effectively combines simulation and real-world experimentation, providing a holistic and well-rounded analysis of the research problem.
- Comparison with SoTA algorithms: The authors show two additional algorithms (Pure Pursuit and MPPI) as a comparison and demonstrate that their RL approach can outperform those in different sections

Weaknesses:
- Currently, the switching of the dynamic models in collisions is unclear "Out of collision, we employ the kinematic bicycle model [37] augmented with data-driven models.113 During collisions, we employ the dynamic bicycle model [13]." -> I do understand that we need correct tyre forces calculations in critical overtaking maneuver, but since the dynamic bycicle model also only has cornering stiffness as a tyre force calculation is should not be to complex. Additionall the authors are writing "If a collision event is detected, the dynamics are switched temporarily to a lower fidelity dynamic model that directly models the effects of external wrenches." --> The dynamic bycicle model is of higher fidelity isn't it? So this sentence needs to be switched?
- The methodological section is quite short and does not capture the concept of the RL with PPO quite good. I can recommend to shortend the Sys-Id Part and Vehicle dynamics model part in order to focus more on the exact method that allows for dynamic overtaking
- Currently, the discussion section of the results is quite poor and does not show in detail

**Quality Of The Limitations Section:**

Limitations are addressed clearly

**Questions For Rebuttal:**

- Why is a team approach something that is worth researching on or better said: Where does the foundation and justification of multi teams is coming from? All-well-known racing series have multiple teams, but they do not cooperate normally (e.g. formula 1). I like the idea but I miss the real-world application ultimately. Furthermore, I would like the authors to explain: What happens if we only train one car without the partner.
- The collision avoidance with the contact wrenches is currently hard to understand and difficult to see, where this is coming from. There are some well-know collision checking models with different fidelity, why have you not chosen those collision checkers e.g. circles or volumes (Sphere, OBB)?
- I suggest to revise Figure 7 completely. I can not see anything in any of those images, both the simulation and real world examples are hard to deciver. It is furthermore unclear how the simulation and the three different types (Overtaking, blocking, Pit) fit together and why we need to see a pit maneuver here. Focus only on the most important parts and elements

**Robotics Focus:**

Sufficient demonstration on hardware

**Summary Of Paper:**

The paper is focusing on autonomous driving with high-speed autonomous ground vehicles. The researchers use an in the community well-known 1:10 scale autonomous vehicle called F1TENTH to demonstrate their research. The goal of the research is to demonstrate wheel-to-wheel racing which means overtaking maneuvers at the limits of handling + close proximity overtaking maneuvers. To achieve that kind of behavior, the authors suggest a learning-based approach which is using self-play reinforcement learning  in simulation to learn compete tive agents with a variety of emergent behaviors. After successful training, the authors demonstrate the results ins simulation and in real-world experiments.

**Summary Of Recommendation:**

In summary, the presented paper shows an interesting approach for autonomous racing and multi-vehicle overtaking and wheel-to wheel maneuvers. The paper has some clear contributions and shows, that their method can be transferred from simulation to real-world autonomous small-scale vehicles. While the manuscript does possess certain strengths and valuable contributions, some areas require further improvements and clarifications. The paper can be improved in several ways e.g. shorten some paragraphs and extend the method section and additional discuss the results more in detail. In summary I think this is a solid conference paper.

---

### Official Review · Reviewer_Edzc · 2023-07-20

**Confidence:** 4
**Originality:** Good
**Technical Quality:** Very Good
**Clarity Of Presentation:** Very Good
**Impact:** 3

**Recommendation:**

Weak Accept: I recommend accepting the paper, but will not argue for my recommendation if the majority of other reviewers have a different opinion.

**Review:**

Overall, I found this paper to be interesting, novel, and clearly written. The problem the authors study is of considerable interest to the robotics community since mixed cooperative-competitive games are quite difficult to solve in general, and have important implications for broader topics such as HRI, autonomous driving, etc.

The hierarchical approach to MARL taken here, while straightforward, does seem to be novel, and I think provides a nice insight for other practitioners seeking to take RL approaches to similar games. Further, the modeling considerations studied by the authors when designing their simulator seem insightful for other researchers considering competitive or high-performance driving problems.

Possible limitations include the limited track geometry shown in the paper, as there are only two track layouts shown in Fig. 7, one simulation, one hardware (will ask a clarifying question about this in the rebuttal and edit as needed). Thus it's unclear how much the resulting policies learn good driving behavior for a specific track, or if the policies generalize to arbitrary track layouts. I will note the observation space is not track-dependent. Similarly, while the agents see a few different driving policies during training (either a heuristic policy or policies from previous checkpoints) it's unclear if this is sufficient to truly "solve" the game in a game theoretic sense (e.g., converge to a Nash equilibrium with new players).

**Quality Of The Limitations Section:**

Additional details required

**Questions For Rebuttal:**

Were the track geometries varied at all during training? Or did agents race repeatedly on the same track? Was the track used in hardware seen during training?

Is it correct to say that the RL-Individual baseline uses the same hierarchical policy archictecture, but the high-level critic doesn't see team-level information when evaluating the Q-function?

**Robotics Focus:**

Sufficient demonstration on hardware

**Summary Of Paper:**

This paper takes a multi-agent RL approach to multi-team car racing. While existing approaches to this problem rely on game theory (which results in slow computation) or "vanilla" multi-agent RL (MARL) (which struggles to learn meaningful team-level behavior), the proposed approach takes a hierarchical strategy, wherein a high-level RL agent takes actions to set team-level goals to be pursued by the low-level agents. The authors develop a lightweight simulator for training that models a number of important dynamic effects (e.g., tire friction and slipping) to enhance sim-to-real transfer. In both simulation and hardware experiments (on 1/10th scale cars), the authors find their method outperforms an MPPI baseline, as well as an ablation where the high-level value function only takes in single-agent state information (a typical CTDE approach).

**Summary Of Recommendation:**

I'd characterize this paper as a weak accept. There are some nice MARL insights here, but this is a crowded area, and, despite a quick lit review, I'm unsure if the work is particularly novel (and would defer to the expertise of other reviewers with more MARL experience). The paper is well-done but I could see it being rejected if the novelty claims are not sufficiently strong.

Edit after rebuttal: After reading the authors' responses, I feel more confident that the approach is novel and would be of interest to the broader robotics community. My score remains a "weak accept" (since this work still skews a bit incremental) but think the paper is of pretty good quality in general.

---

### Official Review · Reviewer_sWPm · 2023-07-20

**Confidence:** 4
**Originality:** Good
**Technical Quality:** Very Good
**Clarity Of Presentation:** Excellent
**Impact:** 3

**Recommendation:**

Weak Accept: I recommend accepting the paper, but will not argue for my recommendation if the majority of other reviewers have a different opinion.

**Review:**

# Strengths

- The agents learn impressive and, as far as I’m aware, novel emergent behavior. Altruistic blocking is particularly impressive given that it is not explicitly rewarded in the reward function and demonstrates an awareness of the importance of team rank, even at the expense of ego-rank and potentially other penalties such as collisions, etc. This suggests that the hierarchical policy formulation and/or reward formulation as designed by the authors is an important contribution and factor of the reported results.

- Domain randomization techniques for facilitating zero-shot transfer are thorough, methodical, and seem to work quite well based off of the reported success rates. The zero-shot transfer and real robot results in impressive and interesting to watch.

# Weaknesses

- The quite complex reward function, despite being arguably the single most important factor of the success rate and emergent behaviors, is only briefly and qualitatively discussed in the main paper and expanded in equation form with little explanation or analysis in the appendix. There are quite a few components in the reward function and my guess is that these were carefully designed and tuned in order to get the sort of emergent interactions we see in the experiments section. The reward function and the design decisions behind it should be analyzed in greater depth. For example, in $r^{HL}$, how important is the $c_1$ term, especially since it is a sparse signal only available at the last timestep? Why is the team rank weighted lower than the ego rank, despite the team rank deciding the outcome of the race? etc.


- **Collisions**. While I understand that certain beneficial team behaviors such as altruistic blocking and cutting-off result in (near) collisions, the reported collision rates are unusually high and seems to be a rather weak spot of the paper. The aforementioned behaviors should still be learnable without necessitating a collision, for example, by inducing a negative reward potential field within a small, fixed radius around each car. The authors could have spent more time investigating and tuning the reward function to address the collision rate as this is important in real F1 and even F1/10 races.

- Figure 3 is difficult to interpret. For one, the y-axis on right-side diagrams are unlabelled. Is this supposed to be a histogram of the frequency rates of various slip angles? Furthermore, it is unclear what information the authors are trying to convey with the figure on the left. Is this supposed to show that the authors have minimized epistemic uncertainty with their model? Why does it seem (at least from this angle) that front steering slip $a_f$ goes down with velocity and left steering? Further analysis and/or a more interpretable diagram would be welcome here.

**Quality Of The Limitations Section:**

Additional details required

**Questions For Rebuttal:**

The velocity limit of 3.5 m/s seems to be limiting, to the point that I'm not sure if the intricate modeling of the front and rear slip angle is relevant. Have the authors already explored the impact of modeling these parameters? An ablation on this would be welcome.

**Robotics Focus:**

Sufficient demonstration on hardware

**Summary Of Paper:**

The authors demonstrate emergent team behaviors in the multi-agent autonomous racing task on real hardware. The contributions can be summarized as:

- A novel, GPU accelerated simulator with various classical and learned dynamics models, and domain randomization to facilitate sim2real transfer

- A hierarchical policy formulation trained with multi-agent reinforcement learning, where the high-level policy attends to long-horizon decision making and goal setting tasks, and the low-level policy attends to low level control

- Demonstrated emergent team behaviors on real hardware

**Summary Of Recommendation:**

The paper is very well written with some surprising results, such as the emergence of certain team behaviors like altruistic blocking. The results are convincing and the demonstrations on hardware are enjoyable to watch. However, I am concerned about the lack of attention and analysis given to the reward function and the impact of the design decision there on the results. I am also concerned about the high collision rate.

---

### Official Review · Reviewer_NBPG · 2023-07-21

**Confidence:** 5
**Originality:** Good
**Technical Quality:** Very Good
**Clarity Of Presentation:** Good
**Impact:** 3

**Recommendation:**

Weak Accept: I recommend accepting the paper, but will not argue for my recommendation if the majority of other reviewers have a different opinion.

**Review:**

The authors propose an interesting challenge in the form of team-based scale car racing. This is relevant for the autonomous driving and wider robotics communities as it combines dynamics and control at various timescales, from team-level strategy to split-second corrections. This pushes the boundaries of what seems easily achievable with existing approaches, e.g. zero-shot sim2real.
Yet the authors also describe a methodology that yields impressive results with interesting emerging behaviour, by using custom accelerated and partially data-driven simulation, self-play, hierarchical control and bilevel optimisation, not to mention actually successfully deploying on hardware. While any single part of the approach in itself is perhaps not incredibly novel, this work gets its value from the well-executed integration and evaluation, and the interesting emerging results.
The main weakness of this submission would have to be the presentation and clarity, which could be improved. Specifically, some figures could use additional annotation and in general the text could use more symbolic notation to describe the ins and outs of various components. E.g. it is not clear how much information agents get from teammates compared to adversaries. Is it just the high-level that sees both? Ideally more details about the training regime would be added to the appendix. More specifics in the question section. The related work could also be improved with the inclusion of recent multi-agent domains in robotics, such as https://arxiv.org/abs/2304.13653.

**Quality Of The Limitations Section:**

Limitations are addressed clearly

**Questions For Rebuttal:**

- Please define "ado" on l61.
- Please elaborate on the difference with [12] based on the results (RL Team vs. RL Independent).
- Section 3.2: please explain why there is a need for a separate in- and out-of-contact models.
- Section 4:
  - Please make more explicit, symbolically, what the exact inputs and outputs of the various components are, and refer to the Appendix for the reward definitions.
  - Please provide more details, possibly in Appendix, about the training regime (batch size, learning rate, etc.) and specifically about opponent sampling.
- Figure 4:
  - Please define "TD".
  - If my understanding is correct, $V_{HL}$ and $\pi_{HL}$ should be $V_{LL}$ and $\pi_{LL}$
- Figure 5: on the radar plots, it is not always immediately clear for which metric which way (higher or lower) is better. Ideally, higher would be better across all metrics.
- Figure 6: the first pair of bars seem to indicate that the win rate of RL Team against MPPI is 100%, which contradicts the 91% mentioned on l230. Please consolidate.
- Figure 7: the top row is very dense and difficult to interpret. If the goal is to show the visualisation of the simulation rather than the behaviour, it might be better to use a single, bigger frame, and add annotations.

**Robotics Focus:**

Sufficient demonstration on hardware

**Summary Of Paper:**

The authors propose a new robotics challenge in the form of a team-based scale care race, where two teams of two cars each race against each other. They implement a zero-shot sim2real strategy, where they implement a GPU-accelerated simulation combining kinematic, dynamic and data-driven models to minimise the transfer gap. Racing policies are trained using self-play and in a hierarchical fashion, where a (team-shared) high-level planner sets position subgoals for a low-level throttle and steering controller to reach. The resulting controllers are compared to a "teamless" RL and MPPI baselines.

**Summary Of Recommendation:**

I vote to accept this paper as it presents both an interesting robotics challenge as well as a well-designed and executed system towards solving it, resulting in interesting emergent behaviour. However as none of the individual parts of the proposed approach are significantly novel, and the merit of the work lies largely in the integration, I struggle to motivate a strong accept.

Edit after rebuttal: following the authors' rebuttal I have increased my recommendation. It is clear from the discussion that the authors have thoroughly investigated the problem, and the additional details in the paper and appendix are very helpful and appreciated. I would even urge the authors to add further details based on the discussions in appendix, e.g. regarding the double model based on contacts.

---

### Decision · Program_Chairs · 2023-08-30

**Decision:**

Accept (Poster)

**Comment:**

Based on the original submission, the rebuttal by the authors and the discussion that followed, the reviewers agree that this paper should be accepted to CoRL.
To the authors: great job on writing a strong rebuttal and engaging with the reviewers during the discussion period! Please address the remaining comments in the camera-ready version of the paper.